# Causality Analysis for COVID-19 among Countries Using Effective Transfer Entropy

**DOI:** 10.3390/e24081115

**Published:** 2022-08-13

**Authors:** Baki Ünal

**Affiliations:** Industrial Engineering Department, Faculty of Engineering and Natural Sciences, İskenderun Technical University, İskenderun 31200, Hatay, Turkey; bakiunal@gmail.com

**Keywords:** COVID-19, causality analysis, causality network, transfer entropy, network analysis

## Abstract

In this study, causalities of COVID-19 across a group of seventy countries are analyzed with effective transfer entropy. To reveal the causalities, a weighted directed network is constructed. In this network, the weights of the links reveal the strength of the causality which is obtained by calculating effective transfer entropies. Transfer entropy has some advantages over other causality evaluation methods. Firstly, transfer entropy can quantify the strength of the causality and secondly it can detect nonlinear causal relationships. After the construction of the causality network, it is analyzed with well-known network analysis methods such as eigenvector centrality, PageRank, and community detection. Eigenvector centrality and PageRank metrics reveal the importance and the centrality of each node country in the network. In community detection, node countries in the network are divided into groups such that countries in each group are much more densely connected.

## 1. Introduction

Investigating causality and information flow between systems is an important area of research in the literature. To evaluate causality and information flow between systems, time series data generated by these systems are utilized. The most widely used causality analysis method in the literature was proposed by Granger [1] and is known as Granger causality. Following Granger, many causality tests are proposed in the literature by authors such as Toda–Yamamoto [2] and Hatemi-J [3] in their own causality tests. Furthermore, in the literature, many nonlinear causality analysis methods are presented [4,5,6,7]; however, all these methods are based on a hypothesis-testing procedure. In these procedures, a test statistic is computed and, according to the value of this statistic, the existence of causality is determined. Therefore, these methods do not measure the strength of causality with a numerical value. A new information-theory-based causality measure, transfer entropy, can measure the strength of the causality [8,9] and can also detect nonlinear causality relationships. In this study, a causality network is constructed using COVID-19 data from seventy countries and effective transfer entropy. The obtained network is a weighted directed network where the directions of links reflect the directions of causality relationships, and the weights of the links indicate the strength of the causalities measured with effective transfer entropy. In the transfer entropy methodology, two kinds of entropy type have been utilized in the literature. These are Shannon entropy and Rényi entropy. Rényi entropy has a free parameter which is denoted by q. When q→1, Rényi entropy converges to Shannon entropy. In this study, we utilized Shannon entropy in our transfer entropy calculations because Shannon entropy is more basic and does not require the selection of a free parameter which complicates the results. In the literature, there are many applications of Rényi entropy-based transfer entropy methodology [10,11,12,13,14]. Ultimately, we chose the transfer entropy methodology for the following reasons: First, unlike other causality assessment methods, transfer entropy not only detects whether there is causality but also measures the strength of this causality. Second, transfer entropy can detect nonlinear relations. Third, in the literature, this methodology is successfully applied to many time series from different fields and its usefulness is proven. Fourth, there is a reliable software for computing transfer entropy. 

After the construction of the causality network, this network is analyzed with network analysis methods such as eigenvector centrality, PageRank, and community detection. Eigenvector centrality and PageRank measure importance and centrality of nodes by assigning a numerical value to each node. In community detection, nodes in the network are divided into groups such that, in each group, the nodes are much more densely connected.

## 2. Materials and Methods

### 2.1. Transfer Entropy

Transfer entropy is an information-theory-based method to quantify information flow and causality between two systems. The concept of transfer entropy was independently formalized by both Thomas Schreiber [8] and Paluš et al. [9]. In the literature, in order to measure the interdependence between two systems, mutual information is purposed by Shannon and Weaver [15]. However, mutual information does not reveal dynamical and directional information. Transfer entropy possess properties of mutual information but also reflects dynamics of information flow (causality). Transfer entropy is based on Shannon entropy. Shannon entropy measures the average number of bits required to encode a discrete random variable, I, possessing the probability distribution p(i) and is expressed with the following formula:(1)$$H_{I}=-\underset{i}{\sum }p\left(i\right)log_{2}p\left(i\right)$$

To obtain optimal encoding based on entropy, the probability distribution p(i) must be known. The amount of bits which will be coded if a different distribution is utilized, such as when q(i) is measured using Kullback entropy, is defined with the following formula [16]:(2)$$K_{I}=-\underset{i}{\sum }p\left(i\right)log_{2}p\left(i\right)/q\left(i\right)$$

Additionally, Kullback entropy for conditional probabilities is expressed with the following formula:(3)$$K_{I|J}=\underset{i,j}{\sum }p\left(i,j\right)log\frac{p(i|j)}{q(i|j)}$$

If the two systems, assumed to be independent, corresponded, then Kullback entropy becomes:(4)$$M_{IJ}=\sum^{\;}p\left(i,j\right)log\frac{p\left(i,j\right)}{p\left(i\right)p\left(j\right)}$$

If transition probabilities are used instead of static probabilities, a dynamic structure can be revealed. For this, it is assumed that the system can be expressed by a Markov process of order k. This means that state $i_{n+1}$ is independent of state $i_{n-k}$. In other words, $p\left(i_{n+1}|i_{n},\ldots ,\;i_{n-k+1}\right)=p\left(i_{n+1}|i_{n},\ldots ,\;i_{n-k+1},\;i_{n-k}\right)$. We note that $i_{n}^{\left(k\right)}=\left(i_{n},\ldots ,\;i_{n-k+1}\right)$. As a result, if the previous states are given an average number of bits required to encode an additional state, it is called the entropy rate and computed with the following formula:(5)$$h_{I}=-\sum^{\;}p\left(i_{n+1},i_{n}^{\left(k\right)}\right)\log\left(i_{n+1},i_{n}^{\left(k\right)}\right)$$

In the expression above, $p\left(i_{n+1},i_{n}^{\left(k\right)}\right)=p\left(i_{n+1}^{\left(k+1\right)}\right)/p\left(i_{n}^{\left(k\right)}\right)$. To analyze information flow between systems, the entropy rate given above is generalized to more than one system by using following expression:(6)$$p\left(i_{n+1},i_{n}^{\left(k\right)}\right)=p(i_{n+1}|i_{n}^{\left(k\right)},j_{n}^{\left(l\right)})$$

According to the expression above, if there is no information flow from J to I, the state of J will not affect the transition probabilities of I. If the expressions above are combined, then transfer entropy can be defined with the following formula:(7)$$T_{J\to I}=\sum^{\;}p\left(i_{n+1},i_{n}^{\left(k\right)},j_{n}^{\left(l\right)}\right)log\frac{p(i_{n+1}|i_{n}^{\left(k\right)},j_{n}^{\left(l\right)})}{p(i_{n+1}|i_{n}^{\left(k\right)})}$$

The transfer entropy calculation described above has a deficiency for small samples. For small samples, the calculated transfer entropies are biased. To solve this problem, the concept of effective transfer entropy is suggested by Marschinski and Kantz [17]. To calculate an effective transfer entropy time, a series of observations from system J are shuffled. From this shuffled data, a transfer entropy is calculated. Then, the transfer entropy obtained from the shuffled data is subtracted from the transfer obtained from the original data. This method can be expressed as follows:(8)$$ET_{J\to I}\left(k,l\right)=T_{J\to I}\left(k,l\right)-T_{J_{Shuffled}\to I}\left(k,l\right)$$

In the expression above, $ET_{J\to I}\left(k,l\right)$ denotes effective transfer entropy and $T_{J_{Shuffled}\to I}\left(k,l\right)$ denotes transfer entropy calculated from shuffled data. With this shuffling procedure, the dependencies in J and between I and J are eliminated. If the sample size is increased, $T_{J_{Shuffled}\to I}\left(k,l\right)$ approaches zero. Therefore, $T_{J_{Shuffled}\to I}\left(k,l\right)$ displays the impact of a small sample size.

The statistical significances of calculated transfer entropies can be evaluated by using a block bootstrap method suggested by Dimpfl and Peter [18]. This method generates the p-values and transfer entropy distribution for the null hypothesis where there is no information flow.

To be used in the calculation of transfer entropy, data should be discrete. Data should be discretized if it is not discrete. To convert a continuous dataset to a discrete dataset, a procedure called symbolic recoding can be used. In this procedure, continuous data are partitioned into bins and each value in the continuous data is assigned to a bin. To perform this procedure, the bounds of the bins should be determined. If these boundaries are determined as $q_{1},q_{2},\ldots ,q_{n}$ $\left(q_{1}<q_{2}<\cdots <q_{n}\right),$ a continuous time series, $y_{t}$, can be made discrete by a symbolic recoding method, as described by following expression.
(9)$$S_{t}=\{\begin{array}{c} \begin{array}{c} 1\;\;\;\;\;\;\;\;\;\;\;\;\;\;\;\;\;\;\;\;for\;y_{t}\leq q_{1}\\ 2\;\;\;\;\;\;\;\;\;\;for\;q_{1}<y_{t}\leq q_{2} \end{array}\\ \vdots \\ \begin{array}{c} n-1\;\;\;\;\;for\;q_{n-1}<y_{t}\leq q_{n}\\ \;\;\;n\;\;\;\;\;\;\;\;\;\;\;\;\;\;\;\;\;\;\;\;\;for\;y_{t}\geq q_{n} \end{array} \end{array}$$

At the end of this procedure, each value in the continuous data is assigned to a number between 1 and *n*.

### 2.2. Network Analysis

In the field of network analysis, several metrics which describe the properties of networks are proposed by Jackson [19] and Newman [20]. In this context, some metrics describe networks’ macrostructure and some metrics describe nodes’ micro properties. Examples of these micro metrics are centrality measures. Centrality measures quantify centrality and the importance of nodes by assigning a value to each node. There are many centrality measures proposed in the literature. Some examples are degree centrality, closeness centrality, betweenness centrality, closeness centrality, eigenvector centrality, and PageRank. In this work, we discuss eigenvector centrality and PageRank. Another network analysis method is community detection. In community detection, nodes of the network are partitioned into communities such that, in each community, the nodes are densely connected.

#### 2.2.1. Eigenvector Centrality

The most basic centrality measure is degree centrality. Degree centrality only takes into account the number of edges that a node has. However, the importance of a node’s neighbors can be different. Eigenvector centrality takes these differences into account. If a node’s neighboring nodes have high centrality (importance), then this node’s centrality (importance) should be high too. Eigenvector centrality is proposed by Bonacich [21]. The eigenvector centrality of a node $x_{i}$ is proportional to the sum of its neighbors’ centralities. Eigenvector centrality can be expressed in the formula below:(10)$$x_{i}=\kappa^{-1}\underset{nodes\;j\;that\;are\;\;neighbors\;of\;i}{\sum }x_{j}$$

In the expression above, $\kappa^{-1}$ is the proportionality constant. The expression above can be rewritten using adjacency matrix $A_{ij}$ of the network, as below:(11)$$x_{i}=\kappa^{-1}\overset{n}{\underset{j=1}{\sum }}A_{ij}x_{j}$$

This expression can be stated in matrix notation, as below:(12)Ax=κx

In the expression above, x is the eigenvector of the adjacency matrix A**,** and its elements are the centrality values.

#### 2.2.2. PageRank

PageRank is a centrality metric which constitutes the core algorithm of Google’s search engine [22]. PageRank is associated with eigenvector centrality and designed for directed networks. To determine the centrality of a node, the PageRank algorithm takes three different factors into account. These are the number of nodes that link to the target, the PageRank centrality of the linking nodes, and the link propensity of the linking nodes. PageRank is calculated with the following formula:(13)$$x={\left(I-\alpha AD^{-1}\right)}^{-1}1$$

In the expression above, α is a positive constant, 1 is the uniform vector (1,1,1,…), D is a diagonal matrix with the elements $D_{ii}=\max\left(k_{i}^{out},1\right)$ ($k_{i}^{out}$ is the outdegree of the node), A is the adjacency matrix, and I is the identity matrix.

#### 2.2.3. Community Detection

Blondel et al. [23] proposed an algorithm for community detection in large networks. This algorithm is also known as the Louvain algorithm. This algorithm consists of two phases, namely the modularity optimization and community aggregation phases. These two phases are iteratively repeated until a convergence. In the first phase, modularity optimization, each node is assigned to a different community. Then, each node, i, and its neighbors, j, are considered, and an evaluation is preformed to determine whether or not modularity will increase if node i is assigned to the community of j. If so, then node i is assigned to the community of j so that the increase in modularity is maximized. This process is repeated until there is no gain in modularity. The increase in modularity when node i is assigned to a community, C, is denoted with ΔQ and computed with the following formula:(14)$$\Delta Q=\left[\frac{\sum_{in}+k_{i,in}}{2m}-{\left(\frac{\sum_{tot}+k_{i}}{2m}\right)}^{2}\right]-\left[\frac{\sum_{in}\;}{2m}-{\left(\frac{\sum_{tot}\;}{2m}\right)}^{2}-{\left(\frac{k_{i}}{2m}\right)}^{2}\right]$$

In the expression above: $\sum_{in}\;$denotes the sum of the weight for links in the community C; $\sum_{tot}\;$denotes the sum of the weight for links in relation to nodes in the community C; $k_{i}$ denotes the sum of the weight for links connected to node i; $k_{i,in}$ denotes the sum of the weight for links from node i to nodes in the community C; and m denotes the sum of the weight for all links in the network.

In the second step, community aggregation, nodes falling in the same community are treated as a single node and a new network is constructed whose nodes comprise the communities from the previous phase. Then, the first step of modularity optimization is executed on the new network. These phases are carried out iteratively until the community structure does not change.

## 3. Results

In this study, we used effective transfer entropy to create a causality network that comprised COVID-19 data from seventy countries. This network is a weighted directed network where the weights of the links correspond to the strength of causality. The links are measured using effective transfer entropy. After constructing the causality network, we analyzed it using network analysis methods such as eigenvector centrality, PageRank, and the Louvain community detection algorithm.

The data used in this study were obtained from the World Health Organization (WHO). The data contain daily new cases for COVID-19 from seventy different countries and cover dates between 3 January 2020 and 7 January 2022 [24]. WHO data for COVID-19 are organized as a table. In this table, the rows indicate days and the columns indicate variables, such as country code, country, new cases, cumulative cases, new deaths, and cumulative deaths. In this study, we use the new cases data from this data table. We present summary statistics for the daily new cases data from each country in Appendix B. Transfer entropy calculations are performed with the RTransferEntropy package of R software [11]. Network analysis and visualization are performed using the Gephi [25] and igraph packages of R software [26]. 

The parameters for transfer entropy calculations are determined as follows: Markov orders for time series are set at one; the number of bootstrap replications used in the evaluation of statistical significance for transfer entropies is set at 300; the number of shuffles used in the calculation of effective transfer entropies is set at 100; and the quantiles used in discretization processes are set at 5% and 95%. For transfer entropy to be applicable, the time series must be stationary. We take the first difference from the daily new cases time series to ensure that the time series are stationary. With augmented Dickey–Fuller tests, we verified that the different time series are stationary.

A graph of the constructed causality network is presented in Figure 1. In this network, only statistically significant transfer entropies are plotted. To improve the visibility, Figure 2 and Figure 3 are presented with some arcs filtered according to their weights. In Figure 2, only the arcs whose weights are greater than 0.05 are plotted, and in Figure 3, only the arcs whose weights are greater than 0.06 are plotted. Therefore, in these figures, only the most important causalities are presented. Arrows showing the directional arcs are better seen in these additional figures. High-definition image files of Figure 1, Figure 2 and Figure 3 and a Pajek network data file of our causality network are provided in the Appendix A. The obtained causality network is a directed weighted network. Directions in the network indicate the directions of causality, and weights in the network reflect the strength of the causality measured with effective transfer entropy. As seen in Figure 1, there is a giant component and an isolated small component in the network. The small component includes the following countries: Egypt, Iraq, Iran, Kuwait, Lebanon, Libya, Pakistan, Qatar, Saudi Arabia, the Syrian Arab Republic, and Tunisia. Notably, these are Islamic countries. There are causalities between these Islamic countries but there are no causality relationships between these Islamic countries and other countries. This is a novel and interesting finding of our study.

After drawing the causality network graph, we investigated the centrality of the countries using eigenvector centrality and PageRank methods. Initially, we calculated eigenvector centrality and PageRank metrics for node countries with Gephi software. However, we found calculated metrics unreliable. For this reason, we recalculated these metrics by using the igraph package of R software. R software version used is 3.6.3. R software is created by R Core Team. R software belongs to R Foundation for Statistical Computing which is settled in Vienna, Austria. R software is sourced from The Comprehensive R Archive Network (cran.r-project.org) web site. In this web site R software is downloaded from a mirror in Denizli, Turkey. Since igraph is more mature than Gephi, igraph is more reliable. The recalculated centrality levels of the countries are presented in Table 1. In Table 1 higher eigenvector centrality and PageRank values correspond to higher centrality and importance of the node countries. We also investigated the community structure in the network. To determine communities, we employed the Louvain algorithm proposed by Blondel et al. [23]. In the causality network, we detected four distinct communities. These communities are presented in Table 2. Countries in each community are much more densely connected.

## 4. Conclusions

In this study, we constructed a causality network for COVID-19 including seventy countries. In this construction, an information-theory-based causality measure, transfer entropy, is utilized. Transfer entropy has some advantages over other causality tests such that it is able to measure the strength of the causality and can detect nonlinear causality relationships. After drawing the causality network, we analyzed it by using centrality measures such as eigenvector centrality and PageRank. We computed the eigenvector centrality and PageRank for each country and presented them in a table. Additionally, we analyzed the community structure in the causality network and detected four distinct communities in the network. 

One finding of our study is that there are two clusters: Islamic countries and other countries. In the causality network, we constructed 1637 directed arcs, and for each arc there is a weight value which reveals the strength of the causality. However, it is impossible for us to present these 1637 weight values in our main text. Therefore, we supplied a Pajek network file in the Appendix A. The Pajek network file we supplied contains 1637 weight values, each corresponding to an arc of the network. Epidemiologists can learn several lessons from our results. For example, in the causality network, if there is a strong causality between one country and another, this means that the first country strongly spread coronavirus to the second country. In this case, the second country could take additional measures against the first, highly contagious country. Epidemiologists can also use data in our causality network to simulate the spread of coronavirus across countries. In our study, we also presented the centrality of countries in the causality network. Countries with high centrality values are hubs for the spread of coronavirus. Therefore, additional measures can be taken in these hub countries to prevent spread of the coronavirus worldwide. Moreover, we detected four communities in our network. These communities reveal strongly connected countries in terms of the spread of coronavirus, and the transmission rates of the coronavirus between these countries is intense. Finally, our results portray the spreading structure of the coronavirus among countries and will be very useful for epidemiologists.

## Figures and Tables

**Figure 1 entropy-24-01115-f001:**
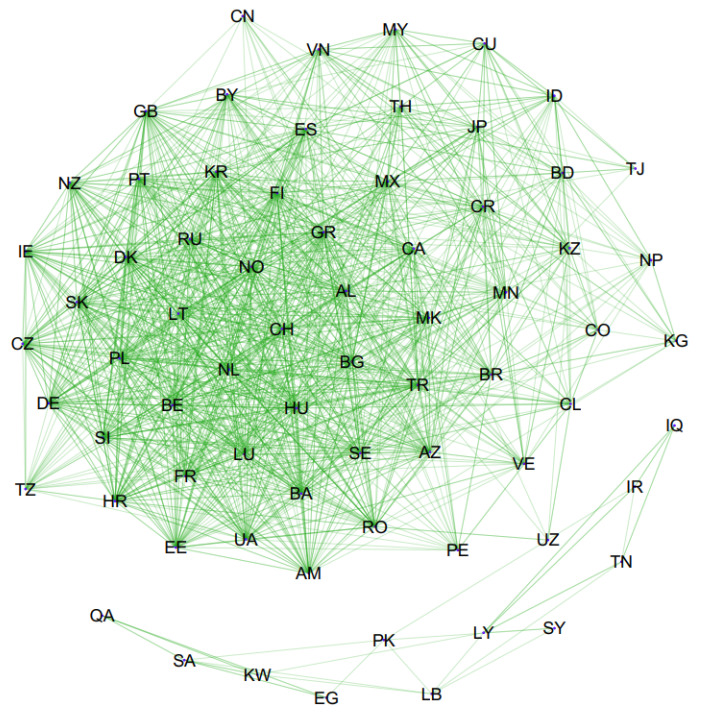
Causality network among seventy countries for COVID-19 using effective transfer entropy.

**Figure 2 entropy-24-01115-f002:**
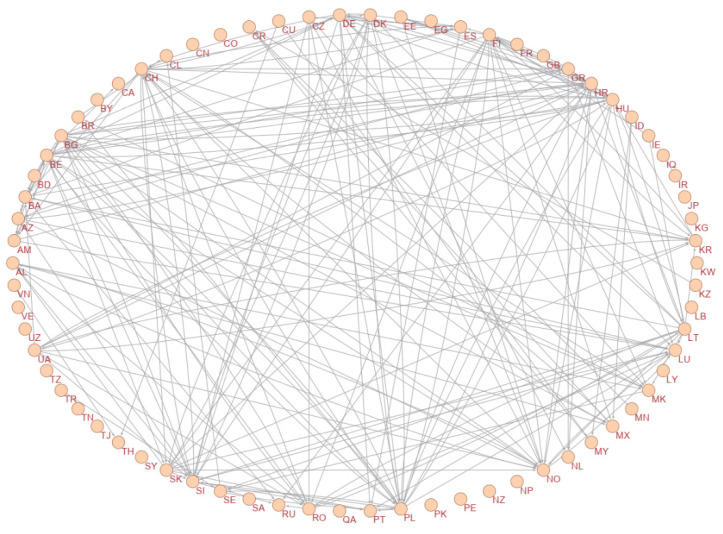
Filtered causality network: Only arcs whose weights are greater than 0.05 are plotted.

**Figure 3 entropy-24-01115-f003:**
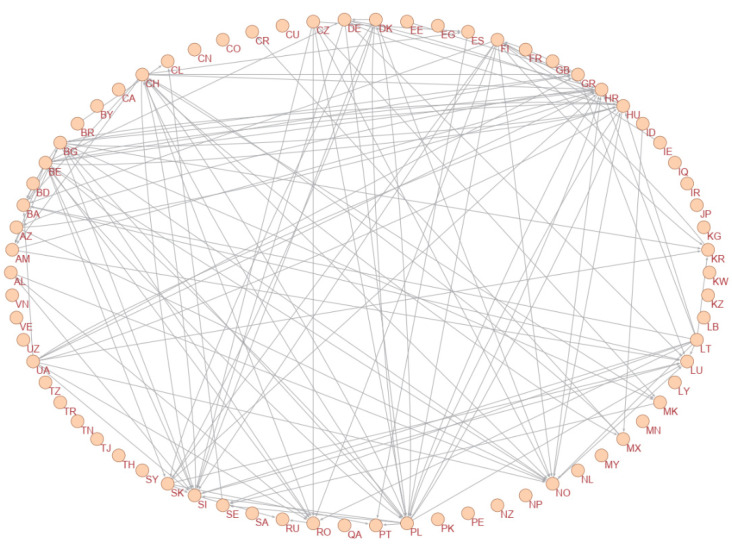
Filtered causality network: Only arcs whose weights are greater than 0.06 are plotted.

**Table 1 entropy-24-01115-t001:** Centrality of node countries.

Country	Eigenvector Centrality	PageRank	Country	Eigenvector Centrality	PageRank
**Albania (AL)**	0.524989	0.022080	**Kuwait (KW)**	0.148186 *	0.008484
**Armenia (AM)**	0.452463	0.014274	**Kazakhstan (KZ)**	0.048409	0.009110
**Azerbaijan (AZ)**	0.522331	0.017451	**Lebanon (LB)**	0.181019 *	0.003974
**Bosnia and Herzegovina (BA)**	0.718292	0.021916	**Lithuania (LT)**	0.474880	0.016001
**Bangladesh (BD)**	0.028208	0.008227	**Luxembourg (LU)**	0.839533	0.026285
**Belgium (BE)**	0.537333	0.017652	**Libya (LY)**	0.628994 *	0.011452
**Bulgaria (BG)**	0.473694	0.016037	**North Macedonia (MK)**	0.384132	0.014058
**Brazil (BR)**	0.200907	0.016291	**Mongolia (MN)**	0.117887	0.010454
**Belarus (BY)**	0.296759	0.014259	**Mexico (MX)**	0.273776	0.015344
**Canada (CA)**	0.288521	0.013702	**Malaysia (MY)**	0.048075	0.010252
**Switzerland (CH)**	0.592722	0.021829	**Netherlands (NL)**	0.465379	0.017194
**Chile (CL)**	0.125363	0.016200	**Norway (NO)**	1.000000	0.033528
**China (CN)**	0.010420	0.003341	**Nepal (NP)**	0.020155	0.005171
**Colombia (CO)**	0.040071	0.005974	**New Zealand (NZ)**	0.253361	0.010018
**Costa Rica (CR)**	0.194700	0.020956	**Peru (PE)**	0.079954	0.006499
**Cuba (CU)**	0.029443	0.009441	**Pakistan (PK)**	0.123575 *	0.007852
**Czechia (CZ)**	0.374990	0.013695	**Poland (PL)**	0.797845	0.024624
**Germany (DE)**	0.584266	0.019286	**Portugal (PT)**	0.495055	0.018178
**Denmark (DK)**	0.472643	0.015801	**Qatar (QA)**	0.088225 *	0.012552
**Estonia (EE)**	0.434803	0.014688	**Romania (RO)**	0.609940	0.020255
**Egypt (EG)**	0.071470 *	0.009371	**Russian Federation (RU)**	0.598568	0.021318
**Spain (ES)**	0.364177	0.017179	**Saudi Arabia (SA)**	0.117345 *	0.013353
**Finland (FI)**	0.578325	0.021833	**Sweden (SE)**	0.518638	0.020728
**France (FR)**	0.287441	0.013241	**Slovenia (SI)**	0.960990	0.029255
**United Kingdom (GB)**	0.252125	0.011740	**Slovakia (SK)**	0.524288	0.018905
**Greece (GR)**	0.634762	0.023944	**Syrian Arab Republic (SY)**	0.144949 *	0.010396
**Croatia (HR)**	0.949647	0.029384	**Thailand (TH)**	0.093885	0.012641
**Hungary (HU)**	0.738304	0.023563	**Tajikistan (TJ)**	0.010324	0.004028
**Indonesia (ID)**	0.029716	0.008124	**Tunisia (TN)**	0.565851	0.008409
**Ireland (IE)**	0.232775	0.009269	**Turkey (TR)**	0.307547	0.014783
**Iraq (IQ)**	1.000000 *	0.012495	**Tanzania (TZ)**	0.059911	0.004068
**Iran (IR)**	0.552977 *	0.009154	**Ukraine (UA)**	0.364791	0.013043
**Japan (JP)**	0.076052	0.009334	**Uzbekistan (UZ)**	0.007412	0.003278
**Kyrgyzstan (KG)**	0.003818	0.004563	**Venezuela (VE)**	0.060501	0.005083
**Korea (KR)**	0.543237	0.023837	**Vietnam (VN)**	0.340229	0.009294

* Eigenvector centrality values for Egypt, Iraq, Iran, Kuwait, Lebanon, Libya, Pakistan, Qatar, Saudi Arabia, Syrian Arab Republic, and Tunisia are calculated from subnetwork.

**Table 2 entropy-24-01115-t002:** Community affiliation of node countries.

Community 1	Community 2	Community 3	Community 4
Armenia (AM)	Albania (AL)	Egypt (EG)	Bangladesh (BD)
Azerbaijan (AZ)	Belarus (BY)	Iraq (IQ)	Brazil (BR)
Bosnia and Herzegovina (BA)	China (CN)	Iran (IR)	Chile (CL)
Belgium (BE)	Czechia (CZ)	Kuwait (KW)	Colombia (CO)
Bulgaria (BG)	Germany (DE)	Lebanon (LB)	Costa Rica (CR)
Canada (CA)	Denmark (DK)	Libya (LY)	Cuba (CU)
Switzerland (CH)	Estonia (EE)	Pakistan (PK)	Indonesia (ID)
Luxembourg (LU)	Spain (ES)	Qatar (QA)	Japan (JP)
North Macedonia (MK)	Finland (FI)	Saudi Arabia (SA)	Kyrgyzstan (KG)
Netherlands (NL)	France (FR)	Syrian Arab Republic (SY)	Kazakhstan (KZ)
Norway (NO)	United Kingdom (GB)	Tunisia (TN)	Mongolia (MN)
Poland (PL)	Greece (GR)		Mexico (MX)
Romania (RO)	Croatia (HR)		Malaysia (MY)
Sweden (SE)	Hungary (HU)		Nepal (NP)
Turkey (TR)	Ireland (IE)		Peru (PE)
Ukraine (UA)	Korea (KR)		Thailand (TH)
Venezuela (VE)	Lithuania (LT)		Tajikistan (TJ)
	New Zealand (NZ)		Uzbekistan (UZ)
	Portugal (PT)		Vietnam (VN)
	Russian Federation (RU)		
	Slovenia (SI)		
	Slovakia (SK)		
	Tanzania (TZ)		

## Data Availability

Data were obtained from the WHO website: https://covid19.who.int/data (accessed on 7 January 2022).

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
