# Peer review of "Causality Analysis for COVID-19 among Countries Using Effective Transfer Entropy"

_entropy, 2022, doi:10.3390/e24081115_

Round 1

Reviewer 1 Report

Baki Ünal: Causality Analysis for COVID-19…, submitted to ENZTROPY 2022

Using WHO COVID data from March, 2021 to June, 2022 the author constructs a causality network for the possible transfer of the disease among seventy countries based on transfer entropy. The theoretical background is shortly described including the most important relations and definitions, the computation performed by means of available standard software packages. The results comprise centrality, page rank  and communities (partially independent groups with internal interdependence) shown in tables and a graphically displayed network showing connections and the existence of communities (only one of them clearly perceivable in the picture).

There are a few points which, in my opinion, must be considered before the article can be published.

1)      The important WHO data are not described in any detail. What do they look like?

2)      The methods applied deliver directed connections (causality!) between countries. The network plot merely displays lines where one expects to find arrows.

3)      There is no legend that explains more about the picture.

4)      There are, in my opinion, two conspicuous and questionable results which deserve a more intensive discussion and, if possible, explanation: the high centrality of Albania and the existence of a community #3 which is connected with the rest of the world only via Uzbekistan. At first side, these results don’t seem very realistic and possibly hint at some problem. Are there other studies which come to similar results? Are the results robust in the sense that they can be reproduced when applying other methods? Is it possible to give an assessment of the underlying data?

5)      The English, although always clear and understandable, can be improved by a natural speaker.

Author Response

Response to Reviewer 1

We thank reviewer for his/her valuable comments. In line with the reviewer's suggestions, we made the necessary additions and changes to our main text.

1.Reviewer Response: The important WHO data are not described in any detail. What do they look like?

1.Authors Response: WHO data for COVID-19 is organized as a table. In this table rows indicate days and columns indicate variables such as country code, country, new cases, cumulative cases, new deaths and cumulative deaths. In this study we used new cases data from this data table. We added these points in our main text.

2.Reviewer Response: The methods applied deliver directed connections (causality!) between countries. The network plot merely displays lines where one expects to find arrows.

2.Authors Response: Our causality network actually contains arrows. But it is hard to see arrows from the Figure 1 in the main text. For this reason, we supplied high resolution vector formatted pdf file of our causality network as “Figure 1. Causality network.pdf” file in compressed file “manuscript-supplementary.zip”. If you zoom to nodes in “Figure 1. Causality network.pdf” file you can see arrows. Also, size of the each arrow is different indicating the difference in the strength of the causalities.

3.Reviewer Response: There is no legend that explains more about the picture.

3.Authors Response: In the Figure 1 there is only nodes which indicates countries and arcs which indicates causality. Since the network is self-explanatory, we didn't need to add a legend. In the network nodes are labelled with ISO country codes which are generally known.

4.Reviewer Response: There are, in my opinion, two conspicuous and questionable results which deserve a more intensive discussion and, if possible, explanation: the high centrality of Albania and the existence of a community #3 which is connected with the rest of the world only via Uzbekistan. At first side, these results don’t seem very realistic and possibly hint at some problem. Are there other studies which come to similar results? Are the results robust in the sense that they can be reproduced when applying other methods? Is it possible to give an assessment of the underlying data?

4.Authors Response: We calculated eigenvector centrality and pagerank values by using Gephi network analysis software. Since results from Gephi is questionable we recalculated eigenvector centrality and pagerank values by using igraph package of the R software. By using values from igraph package we reconstructed Table 1. Since Egypt, Iraq, Iran, Kuwait, Lebanon, Libya, Pakistan, Qatar, Saudi Arabia, Syrian Arab Republic and Tunisia are in an isolated component in the graph when whole network is used eigenvector centrality of these countries are calculated as zero. To overcome this problem eigenvector centrality values for these countries are computed from subnetwork formed by these countries.

5.Reviewer Response: The English, although always clear and understandable, can be improved by a natural speaker.

5.Authors Response: We checked the English of our paper.

Reviewer 2 Report

Referee report on: “Causality Analysis For COVID-19 Among Countries Using Effective Transfer Entropy” (entropy-1825072)

This paper relies on the concept of transfer entropy to study the causality relationships among various countries in terms of the number of daily new COVID-19 cases. The method is applied on 70 countries during the 2020-2022 time period and it identified two distinct clusters of countries, but no further economic or health insights have been provided.    

Comments:

1.      The paper’s topic is an important one. However, there are quite a few theoretical and empirical considerations that it does not cover sufficiently well. First, why was transfer entropy your only choice? Is it the “optimal” entropy type for this application? I would like to see at least a single competing analysis that involves an alternative entropy. Please see, for instance, Stutzer, M. The Role of Entropy in Estimating Financial Network Default Impact. Entropy 2018, 20, 369. https://doi.org/10.3390/e20050369

2.      What is the relationship between your method and a non-linear causality analysis in the spirit of Lento, C.; Gradojevic, N. S&P 500 Index Price Spillovers around the COVID-19 Market Meltdown. J. Risk Financial Manag. 2021, 14, 330. https://doi.org/10.3390/jrfm14070330. Is your method more accurate and/or more beneficial? Please discuss.

3.      Data: We need a summary stats table for the daily new cases and potentially some sub-period analysis. Is this a stationary time series for each country?

4.      My understanding is that the whole data set is used to construct transfer entropies. It would be more interesting to observe how the network causalities changed during the course of the pandemic from 2020 to 2022. Hence, a more detailed temporal analysis is a must.

5.      Finally, the results are not very informative. We learn that there are two causality clusters, Islamic and other countries, but the authors do not provide any economic intuition or explanation for their results. What is going on here? What are the lessons for the epidemiologists? Could we learn something about the economic ties of the Islamic vs. non-Islamic countries during the pandemic? The paper should have more layers of results that are tied together into a sensible story. In my opinion, the paper could generate more intriguing results if my recommendations above are followed with regards to alternative entropies, temporal analysis and careful interpretation of results that clearly shows where the current paper stands relative to the literature.

Author Response

Response to Reviewer 2

We thank reviewer for his/her valuable comments. In line with the reviewer's suggestions, we made the necessary additions and changes to our main text.

1.Reviewer Response: The paper’s topic is an important one. However, there are quite a few theoretical and empirical considerations that it does not cover sufficiently well. First, why was transfer entropy your only choice? Is it the “optimal” entropy type for this application? I would like to see at least a single competing analysis that involves an alternative entropy. Please see, for instance, Stutzer, M. The Role of Entropy in Estimating Financial Network Default Impact. Entropy 2018, 20, 369. https://doi.org/10.3390/e20050369

1.Authors Response: The most widely used causality analysis method in the literature was proposed by Granger (1969) and known as Granger Causality. Following Granger many causality tests are proposed in the literature such Toda-Yamamoto (1995) and Hatemi-J (2012) causality tests. However, most of these causality tests only indicate whether there is a causality or not and they do not measure the strength of causality. We choose transfer entropy to analyze causality between daily new cases series from seventy countries. In the transfer entropy methodology two kinds of entropy type utilized in the literature. These are Shannon entropy and Rényi entropy. Rényi entropy has a free parameter which is denoted by q. When q1 Rényi entropy converges to Shannon entropy. In this study we utilized Shannon entropy in our transfer entropy calculations because Shannon entropy is more basic and do not require selection of a free parameter which complicates the results. In the future studies causality for COVID-19 can be investigated with Rényi entropy based transfer entropy methodology. Example applications of Rényi transfer entropy are Behrendt et al. (2019), He & Shang (2017), Assaf et al. (2022), Adam (2020), Jizba et al. (2021). Ultimately, we chose transfer entropy methodology for the following reasons: Firstly, unlike other causality assessment methods transfer entropy not only detects whether there is causality, but also measures the strength of this causality. Secondly, transfer entropy can detect nonlinear relations. Thirdly, in the literature this methodology is successfully applied to many time series from different fields and its usefulness is proven. Fourthly, there is a reliable software for computing transfer entropy. We added these points to our main text.

Granger, C. W. (1969). Investigating causal relations by econometric models and cross-spectral methods. Econometrica: Journal of the Econometric Society, 37, 424-438.

Toda, H. Y., & Yamamoto, T. (1995). Statistical inference in vector autoregressions with possibly integrated processes. Journal of Econometrics, 66(1-2), 225-250.

Hatemi-j, A. (2012). Asymmetric causality tests with an application. Empirical Economics, 43(1), 447-456.

Behrendt, S., Dimpfl, T., Peter, F. J., & Zimmermann, D. J. (2019). RTransferEntropy—Quantifying information flow between different time series using effective transfer entropy. SoftwareX, 10, 100265.

He, J., & Shang, P. (2017). Comparison of transfer entropy methods for financial time series. Physica A: Statistical Mechanics and its Applications, 482, 772-785.

Assaf, A., Bilgin, M. H., & Demir, E. (2022). Using transfer entropy to measure information flows between cryptocurrencies. Physica A: Statistical Mechanics and its Applications, 586, 126484.

Adam, A. M. (2020). Susceptibility of stock market returns to international economic policy: evidence from effective transfer entropy of Africa with the implication for open innovation. Journal of Open Innovation: Technology, Market, and Complexity, 6(3), 71.

Jizba, P., Lavička, H., & Tabachová, Z. (2021). Rényi Transfer Entropy Estimators for Financial Time Series. Engineering Proceedings, 5(1), 33.

2.Reviewer Response: What is the relationship between your method and a non-linear causality analysis in the spirit of Lento, C.; Gradojevic, N. S&P 500 Index Price Spillovers around the COVID-19 Market Meltdown. J. Risk Financial Manag. 2021, 14, 330. https://doi.org/10.3390/jrfm14070330. Is your method more accurate and/or more beneficial? Please discuss.

2.Authors Response: In the literature many non-linear causality analysis methods are presented (Lento et al., 2021; Breitung & Candelon, 2006; Hiemstra & Jones, 1994; Diks & Panchenko, 2006). However, all these methods are based on a hypothesis testing procedure. In these procedures a test statistic is computed and according to value of this statistic existence of causality is determined. Therefore, these methods do not measure the strength of causality with a numerical value. However, transfer entropy based on different methodology and can measure the strength of a causality by a numerical value. We added these points to our main text.

Lento, C., & Gradojevic, N. (2021). S&P 500 index price spillovers around the COVID-19 market meltdown. Journal of Risk and Financial Management, 14(7), 330.

Breitung, J., & Candelon, B. (2006). Testing for short-and long-run causality: A frequency-domain approach. Journal of econometrics, 132(2), 363-378.

Hiemstra, C., & Jones, J. D. (1994). Testing for linear and nonlinear Granger causality in the stock price‐volume relation. The Journal of Finance, 49(5), 1639-1664.

Diks, C., & Panchenko, V. (2006). A new statistic and practical guidelines for nonparametric Granger causality testing. Journal of Economic Dynamics and Control, 30(9-10), 1647-1669.

3.Reviewer Response: Data: We need a summary stats table for the daily new cases and potentially some sub-period analysis. Is this a stationary time series for each country?

3.Authors Response: We added summary statistics table in our main text. As we mentioned in our main text, we take first difference of time series for each country. And we verified that all differenced series are stationary by using augmented Dickey-Fuller test.

4.Reviewer Response: My understanding is that the whole data set is used to construct transfer entropies. It would be more interesting to observe how the network causalities changed during the course of the pandemic from 2020 to 2022. Hence, a more detailed temporal analysis is a must.

4.Authors Response: Like many statistical methods, in the transfer entropy when the data is short, small sample bias arise. In our analysis our data contains only 735 observations (days). This data is relatively short. Therefore, we did not carry out temporal analysis such as sub-period or rolling window analysis.

5.Reviewer Response: Finally, the results are not very informative. We learn that there are two causality clusters, Islamic and other countries, but the authors do not provide any economic intuition or explanation for their results. What is going on here? What are the lessons for the epidemiologists? Could we learn something about the economic ties of the Islamic vs. non-Islamic countries during the pandemic? The paper should have more layers of results that are tied together into a sensible story. In my opinion, the paper could generate more intriguing results if my recommendations above are followed with regards to alternative entropies, temporal analysis and careful interpretation of results that clearly shows where the current paper stands relative to the literature.

5.Authors Response: One finding of our study is that there are two clusters: Islamic and other countries. In our study we constructed a causality network which reveals causalities and strength of causalities. In our causality network we have 1637 directed arcs. And for each arc there is e weight value which reveals the strength of the causality. However, it is impossible for us to present these 1637 weight values in our main text. Therefore, we supplied pajek network file namely “Pajek Network File of the Causality Network.net” in the compressed file “manuscript-supplementary.zip”. Pajek network file we supplied contains 1637 weight values of each arc of the networks. Epidemiologists can learn several lessons from our results. For example, in the causality network if there is a strong causality from one country to another this means that first country strongly spreads coronavirus to second country. In this case, in the second country additional measures can be taken against the first highly contagious country. Epidemiologists can also use data in our causality network to simulate spread of coronavirus across countries. In our study we also presented centrality of countries in the causality network. Countries with high centrality values are hubs for the spread of coronavirus. Therefore, additional measures can be taken in these hub countries to prevent spread of the coronavirus worldwide. Also, we detected four communities in our network. These communities reveal strongly connected countries in terms of spread of coronavirus and the transmission of the coronavirus between these countries is intense. Eventually our results portray the spreading structure of the coronavirus among countries and will be very useful for epidemiologists. We added these points to our main text.

Round 2

Reviewer 1 Report

Baki Ünal: Causality Analysis for COVID-19…, submitted to ENZTROPY 2022

The interesting paper was improved by describing WHO data and applying a second method called igraph for analyzing the computed network. The centrality values have changed dramatically and now seem to be more reasonable.

There are still three points which, in my opinion, must be considered before the article can be published.

1)      The transition from Gephi to igraph software (a) and the separate treatment of the Islamic state  group (b) are essential for the new, more reasonable centrality values listed in Table 1. For the sake of scientific clarity I propose to mention and discuss the new approach. If Gephi is really questionable as claimed in the response to my points, then this is valuable information for others. 

2)      What was more important, step (a) or step (b)?

3)      In Figure 1, causality is displayed as an arrow pointing from state “A” to state “B”. Unfortunately, however, the arrow can be identified only for a few pairs of states with a low number of connections to others. This is true for the group of Islamic states. Other states are surrounded by a high number of arrowheads that cannot be attributed to a particular connection line. In addition, the state labels are often hiding arrowgeads. So, for the majority of pairs the direction of the arrow cannot be understood from the figure and the interesting information (the major outcome of the article) is not available. What is the solution of the problem? A simplified figure containing only the most important arrows with causalities above some lower limit will certainly be useful for a fast overview. Please check if arrowheads are all visible. Maybe also a complete list can make sense as supplementary information, but it can also be omitted.

Author Response

Response to Reviewer 1

We thank reviewer for his/her valuable comments. In line with the reviewer's suggestions, we made the necessary additions and changes to our main text.

1.Reviewer Response: The transition from Gephi to igraph software (a) and the separate treatment of the Islamic state group (b) are essential for the new, more reasonable centrality values listed in Table 1. For the sake of scientific clarity, I propose to mention and discuss the new approach. If Gephi is really questionable as claimed in the response to my points, then this is valuable information for others.

1.Authors Response: Both Gephi and igraph are open-source software. (In fact, igraph is not a standalone software but a library which is available for R, Python, Mathematica and C). Latest version of Gephi is 0.9.7 and latest version of igraph for R is 1.3.4. Since igraph is more mature than Gephi, igraph is more reliable. Different outcomes from Gephi and igraph must be result of their algorithmic implementations. Therefore, the source codes of these two software should be examined. We reported the problem we encountered to the developers of Gephi. We mentioned these points in our main text.

2.Reviewer Response: What was more important, step (a) or step (b)?

2.Authors Response: The transition from Gephi to igraph software (step (a)) is more important.

3.Reviewer Response: In Figure 1, causality is displayed as an arrow pointing from state “A” to state “B”. Unfortunately, however, the arrow can be identified only for a few pairs of states with a low number of connections to others. This is true for the group of Islamic states. Other states are surrounded by a high number of arrowheads that cannot be attributed to a particular connection line. In addition, the state labels are often hiding arrowheads. So, for the majority of pairs the direction of the arrow cannot be understood from the figure and the interesting information (the major outcome of the article) is not available. What is the solution of the problem? A simplified figure containing only the most important arrows with causalities above some lower limit will certainly be useful for a fast overview. Please check if arrowheads are all visible. Maybe also a complete list can make sense as supplementary information, but it can also be omitted.

3.Authors Response: In line with the reviewer’s recommendations, to improve visibility, we presented two additional filtered network graph in Figure 2 and Figure 3. These figures are obtained by removing arcs whose weights are lower than specific values. In Figure 2 only arcs whose weights are greater than 0.05 are plotted. And In Figure 3 only arcs whose weights are greater than 0.06 are plotted. Arrows are better seen in these additional figures. We also supplied high resolution eps files of Figure 2 and Figure 3 in supplementary data section.

Reviewer 2 Report

OK, thank you. Congrats on the publication in Entropy!

Author Response

Response to Reviewer 2

Thank you very much.